# The Technique and Advantages of Contrast-Enhanced Ultrasound in the Diagnosis and Follow-Up of Traumatic Abdomen Solid Organ Injuries

**DOI:** 10.3390/diagnostics12020435

**Published:** 2022-02-08

**Authors:** Marco Di Serafino, Francesca Iacobellis, Maria Laura Schillirò, Roberto Ronza, Francesco Verde, Dario Grimaldi, Giuseppina Dell’Aversano Orabona, Martina Caruso, Vittorio Sabatino, Chiara Rinaldo, Luigia Romano

**Affiliations:** Department of General and Emergency Radiology “Antonio Cardarelli” Hospital, 80131 Naples, Italy; iacobellisf@gmail.com (F.I.); marialaura.schilliro@gmail.com (M.L.S.); roberto.ronza@hotmail.it (R.R.); francescoverde87@gmail.com (F.V.); dariogrimaldi@me.com (D.G.); giuseppinadellaversanoorabona@gmail.com (G.D.O.); caruso.martina90@gmail.com (M.C.); vittorio.sabatino@gmail.com (V.S.); chiara_rinaldo@libero.it (C.R.); luigia.romano1@gmail.com (L.R.)

**Keywords:** CEUS, blunt trauma, non-operative management, follow-up

## Abstract

Trauma is one of the most common causes of death or permanent disability in young people, so a timely diagnostic approach is crucial. In polytrauma patients, CEUS (contrast enhanced ultrasound) has been shown to be more sensitive than US (ultrasound) for the detection of solid organ injuries, improving the identification and grading of traumatic abdominal lesions with levels of sensitivity and specificity similar to those seen with MDCT (multidetector tomography). CEUS is recommended for the diagnostic evaluation of hemodynamically stable patients with isolated blunt moderate-energy abdominal traumas and the diagnostic follow-up of conservatively managed abdominal traumas. In this pictorial review, we illustrate the advantages and disadvantages of CEUS and the procedure details with tips and tricks during the investigation of blunt moderate-energy abdominal trauma as well as during follow-up in non-operative management.

## 1. Introduction

Trauma is one of the most common causes of death or permanent disability in young people, so a timely diagnostic approach is crucial. Multidetector computed tomography (MDCT) with intravenous iodinated contrast is excellent at detecting and characterizing life-threatening injuries. Its use is crucial in the initial assessment of polytraumatized patients to determine whether a surgical, interventional, or non-operative treatment approach is best [1,2,3]. Given the need to reduce exposure to ionizing radiation and to consider the risk of contrast-induced nephropathy, the appropriate selection of trauma patients for MDCT is becoming more critical [4,5]. Furthermore, in recent years a more conservative approach for traumatic injuries of parenchymatous organs has been encouraged; at present, non-operative management is considered the standard treatment [6,7]. Consequently, this has led to an increased number of imaging studies to monitor the healing of lesions [8,9]. Due to the invasiveness associated with the use of intravenous contrast medium and ionizing radiation, MDCT is not the preferred follow-up imaging method for hemodynamically stable patients or isolated, blunt, moderate-energy abdominal trauma. Instead, at present, there is a trend toward the use of less invasive imaging methods, such as contrast-enhanced ultrasound (CEUS) [8]. In polytrauma patients, CEUS has been shown to be more sensitive than ultrasound (US) in the detection of solid organ injuries. CEUS is able to identify and grade traumatic abdominal lesions with sensitivity and specificity levels similar to those seen in MDCT, which reach up to 95% [10]. CEUS is a radiation-free technique with good diagnostic accuracy for identifying parenchymal and vascular injuries [11,12]. According to EFSUMB (European Federation of Societies for Ultrasound in Medicine and Biology) guidelines, CEUS can be used as an alternative to computer tomography (CT) scans for hemodynamically stable patients with isolated, blunt, moderate-energy abdominal traumas to evaluate solid organ injury, particularly for children and for follow-ups of conservatively managed abdominal trauma to reduce the number of CT examinations [13,14]. Furthermore, CEUS has many other applications in studying parenchymatous and vessel non-traumatic pathologies [15,16,17,18,19,20,21,22,23] and as a problem solving technique [24], and also in guiding abdominal interventional procedures [25].

## 2. Findings and Procedure Details

### 2.1. Instrumentation

#### 2.1.1. Ultrasound System with Contrast Imaging Package Software

The US transducer should operate at a low mechanical index (MI), generally below 0.3. Additionally, it should be able to analyze the resonance signals originated by the contrast agent while avoiding the destruction of the bubble and reducing tissue harmonics and artifacts [13,26].

#### 2.1.2. Ultrasound Contrast Agent

The ultrasound contrast agents (UCAs) currently approved in Europe are sulfur hexafluoride microbubbles (SonoVue™, Bracco, Milan, Italy) and perflutren microspheres (Optison, GE Healthcare™ and Luminity™, Lantheus Medical Imaging, North Billerica, MA, USA). From our experience, we adopted SonoVue; it is a pure intravascular agent consisting of micro-bubbles (1–7 micron) that contain sulfur hexafluoride encapsulated by a phospholipid shell. Micro-bubbles are too large to pass through the vascular endothelium and stay intact for up to 7 min in the blood vessels. After they dissolve, the gas is exhaled through the lungs, and the phospholipid shell is metabolized, primarily in the liver. There is no need for blood tests before UCA injection as the agent is not excreted by the kidneys, rather through the lungs during breathing. Therefore, renal insufficiency is not a contraindication for UCA injection as there is no risk of contrast-related nephropathy or nephrogenic systemic fibrosis associated with their use [4,27]. Furthermore, there is no evidence of any effect on thyroid function because UCAs do not contain iodine. UCAs are generally well tolerated; the rate of adverse reactions is very low (1:7000 patients, 0.014%). This rate is significantly lower than the rate associated with iodinated state-of-the-art CT agents (35–95:100,000 patients, 0.035–0.095%) [13,28]. The main contraindications for UCAs are a history of allergic reactions to the contrast agent, severe pulmonary hypertension, severe coronary artery disease, pulmonary hypertension, and unstable ischemic heart disease. The use of UCAs is not authorized for pregnant or breastfeeding women [13,28]. The use of UCAs is yet to be off-label for children; however, there is a large consensus on their safety [13,29].

#### 2.1.3. Needle with a Diameter of at Least 32 Gauge

It is important to avoid the rupture of micro-bubbles under injection pressure [13,30].

### 2.2. Procedure Details

Examination starts with the non-enhanced US. All parenchymatous organs and the peritoneal cavity are investigated to determine the presence of parenchymatous injuries that would need to be deeply studied following injection of the UCA. As there is limited time to scan each organ following the injection due to the timing of each vascular phase, it is important to find areas requiring further investigation before the injection is administered (Figure 1) [30].

The administration of the UCA should be preceded by a preliminary study with a color and power Doppler US (CD–US) of the injured parenchyma to identify any contained vascular lesions. This increases the diagnostic confidence in differentiating these lesions through their characteristic spectral pulsed-wave Doppler (Figure 2).

Indeed, after UCA administration, these lesions could become indeterminate at CD–US evaluation due to the relative turbulence generated by the micro-bubbles; furthermore, the relative destruction of the micro-bubbles due to high MI during the Doppler study may also render the contrast study ineffective (Figure 3).

CEUS examination is performed after intravenous administration of a 2 mL bolus of UCA (90 μg of sulfur hexafluoride), followed by approximately 10 mL of saline solution administered through an antecubital vein.

The US software can host a split-screen to enable the baseline greyscale B-mode image as reference images while the tissue enhancement is evaluated through the low MI CEUS [13]. Like MDCT, CEUS enables the evaluation of all contrastographic phases, particularly the early arterial phase. CEUS can also continuously scan the region of interest during each contrast phase. The arterial phase starts at 10–20 s and continues for up to 30–40 s following contrast injection. The advantage of scanning organs in the early arterial phase is obtaining the optimal depiction of contained vascular injuries, such as pseudoaneurysms and arteriovenous fistulas, that may appear days after the trauma occurred. Such injuries may be responsible for late organ rupture or, in later stages, the alteration of the systemic circulation. 

During the venous and late phases, which occur 2–6 min following injection, the contrast agent is distributed to the entire capillary bed. 

Flash mode, a technique specific to CEUS-capable US devices, emits a short US pulse with very high MI to destroy accumulated micro-bubbles within an area of interest. This enables the re-evaluation of dynamic post-contrast perfusion, as long as the concentration of the UCA slowly decreases prior to its excretion through the lungs (Table 1) [13].

The timing of parenchymal enhancement after intravenous administration of the UCA depends on the vascular anatomical and physiological differences in each organ and on the hemodynamic status of the patient. For example, this timing in elderly patients may be influenced by their cardiac function [30].

Kidneys show the most rapid parenchymal enhancement, followed by the liver, pancreas, and adrenal glands, which show intermediate enhancement patterns. The spleen has a later and more persistent enhancement compared to the kidneys. The timing of these enhancements is important if multiple organs require imaging during the same examination (Table 2).

**Kidney**: the cortex enhances quickly and intensely after the injection, while the pyramids enhance from the periphery to the center in approximately 30 s [13,26]. The optimal time window for renal parenchymal injury assessment is up to 2.5 min following injection, as this is when maximum enhancement of the kidney can be observed [27,30] (Figure 4). 

UCA does not accumulate in the pelvicalyceal system; therefore, no excretory phase will occur. 

**Liver**: the arterial phase starts between 10–40 s after the injection. The hepatic and portal phases begin between 40–120 s after the injection, while the sinusoidal phase begins between 120–300 s after. Due to the dual vascular supply in the liver, a homogeneous parenchymal enhancement is shown that is adequate for the detection of organ injury (Figure 5) [13,26,30].

**Pancreas**: the perfusion of the pancreas occurs early and intensely, with progressive wash-out. The arterial phase occurs 15–30 s following injection. This phase is recognizable by direct visualization of the aorta, celiac tripod, and superior mesenteric artery, which immediately precede the enhancement of the parenchymal gland. The venous phase occurs 30–120 s after injection and is associated with the visualization of the splenic-portal axis. This phase corresponds to the homogeneous enhancement of the parenchyma gland, which is adequate for the detection of organ injury [13,26,30,31] (Figure 6).

**Adrenal glands**: the arterial phase starts 20–40 s after the injection, followed by a venous phase where a progressive enhancement of the parenchyma gland is observed for up to 5 min. This enhancement is adequate for organ injury detection [13,26,30]. 

**Spleen**: the arterial phase starts 12–20 s after the injection. This phase shows irregular enhancement, similar to what is seen during MDCT, making it difficult to define any parenchymal injury. The venous phase starts 40–60 s after the injection. This phase provides adequate organ injury detection as the healthy parenchyma appears with a homogeneous enhancement for 5–7 min (Figure 7) [13,26,30].

#### 2.2.1. Haemodynamically Stable Patients with Isolated Blunt Moderate-Energy Abdominal Trauma

The examination should begin with the kidneys during the arterial phase because their enhancement occurs quickly and is fleeting [13]. Following kidney examination, the adrenal glands, liver, pancreas, and spleen should be evaluated during the venous phase, as the possible area of parenchymal laceration is better highlighted in this phase. The CEUS examination typically utilizes two split doses of intravenous UCA, one for each side of the body. More specifically, one dose is used to evaluate the right kidney, right adrenal gland, liver, and pancreas, while the second dose is used to assess the left kidney, left adrenal gland, and spleen [13]. Once exploration of all abdominal parenchyma and identification of areas containing parenchymal injuries is complete, injured areas should be re-evaluated under flash mode with the same dose of UCA. This is so the entire dynamic contrastographic study of the injured area can be observed, and any vascular lesions can be excluded. Any plurifocal or multi-organ parenchymal injuries, active bleeding, or contained vascular injuries still require diagnostic investigation with MDCT examination using intravenous iodinated contrast medium to understand the best therapeutic strategy to treat such injuries with more excellent study overview (Figure 8).

#### 2.2.2. Follow-Up of Conservatively Managed Abdominal Trauma

In follow-up CEUS examinations, the known injured organ is targeted, and all contrastographic phases are evaluated to exclude any contained vascular lesions in the arterial phase. Any regression of the parenchymal injured area is monitored during the venous and late phases [13]. In the event of any worsening changes in the post-traumatic findings, the use of MDCT with intravenous iodinated contrast medium administration is mandatory for the same reasons as above (Figure 9).

### 2.3. Findings

Imaging findings depend on the contrast media distribution. In normal parenchyma, the distribution is homogeneous with a clear depiction of the vascular structures. CEUS may accurately define organ injuries, capsular extensions, and vascular injuries with accuracy similar to MDCT [12,13,30,32].

#### 2.3.1. Solid Organ Injuries May Involve the Parenchyma and the Vessel

Parenchymal injuries:

Intraparenchymal haematoma: the haematoma appears as a focal non-enhancing elliptic collection in the parenchyma with poorly defined irregular margins and no internal enhancing vessels. It does not involve interruption of the organ capsule and is particularly evident during the venous phase of the study (Figure 10 and Figure 11) [12,30].

Lacerations: these findings are identifiable as irregular linear or branched non-enhancing bands, frequently perpendicular to the organ capsule, and can be associated with capsular discontinuity. Lacerations can be classified as superficial (≤3 cm in depth) or deep (>3 cm in depth) (Figure 12, Figure 13, Figure 14, Figure 15 and Figure 16) [13]. 

#### 2.3.2. Vascular Injuries

Active bleeding:

Active bleeding can be observed as micro-bubble extravasation outside blood vessels within the peritoneal or retroperitoneal space (Figure 17) [1,30].

Contained vascular injuries:

Contained vascular injuries include pseudoaneurysms and arteriovenous fistulas. Pseudoaneurysms are focal outpouchings of the external vessel contour due to the partial disruption of the wall, which is contained by the tissue around the vessel (Figure 18, Figure 19 and Figure 20) [1,30]. 

Arteriovenous fistulas consist of traumatic communication between the arterial and venous systems [1,30]. Fistulas are characterized as asymmetrical, early contrast opacification of a vein during the early arterial phase of the study (Figure 21).

## 3. Advantages and Disadvantages

MDCT with intravenous iodinated contrast medium is the primary imaging method used for total body evaluation in high-energy blunt trauma patients. MDCT can provide precise delineation of a parenchymal laceration or contusion, indicating the presence of devascularized segments and vascular lesions [33]. However, the use of ionizing radiation and intravenous iodinate contrast medium are limiting factors, particularly for injuries with a moderate risk mechanism and for follow-ups of conservatively managed abdominal traumas. For the latter, recourse to CEUS is strongly recommended as CEUS can achieve 99% sensitivity and specificity, avoiding overutilization of CT [13].

CEUS is fast, cheap, simple to perform, and can be performed bedside. It does not use ionizing radiation, and USCAs are well tolerated; anaphylactoid reactions to these agents are rare. Furthermore, there are few contraindications for its use [30,34,35,36]. 

The limitations of CEUS are the possibility that adequate exploration of organs of interest may be obscured by bowel gas interposition or due to the body habitus. For example, the pancreas can be hidden due to its deep retroperitoneal location. CEUS cannot diagnose traumatic lesions of the pelvicalyceal system as the kidneys do not excrete micro-bubbles. Likewise, CEUS cannot diagnose bile duct injuries with bile leakage because micro-bubbles are not eliminated by the biliary tract [8,13,30]. Furthermore, patients with bowel injuries, high-energy blunt traumas, multi-organ trauma, or those who are hemodynamically unstable should undergo MDCT examination rather than CEUS examination. 

## 4. Case Series: Step-by-Step Practical Applications, Tips and Tricks during CEUS Follow-Up of Conservatively Managed Abdominal Trauma

### 4.1. Step 1

Choose the imaging in your strings (Figure 22).

### 4.2. Step 2

“Turn on the light”: CEUS is more sensitive than US for the detection of solid organ injuries (Figure 23).

### 4.3. Step 3

Beware of vascular injuries during follow-up of parenchymal injuries (Figure 24 and Figure 25).

### 4.4. Step 4

Beware of lacerated areas of the parenchyma (pseudo-nodular spared area). Always integrate with a preliminary CD–US study before CEUS examination (Figure 26).

### 4.5. Step 5

Beware to the growing parenchymal collections in the suspicion of a vascular, biliary, or urinary leak (Figure 27).

## 5. Conclusions

In polytrauma patients, CEUS has been shown to be more sensitive than US for the detection of solid organ injuries, improving the identification and grading of traumatic abdominal lesions with levels of sensitivity and specificity similar to those seen with MDCT. CEUS is recommended for the diagnostic evaluation of hemodynamically stable patients with isolated blunt moderate-energy abdominal traumas and for the diagnostic follow-up of conservatively managed abdominal traumas. Using CEUS minimizes inappropriate or further exposure to ionizing radiation, and multiple intravenous iodinate medium contrast administration.

## Figures and Tables

**Figure 1 diagnostics-12-00435-f001:**
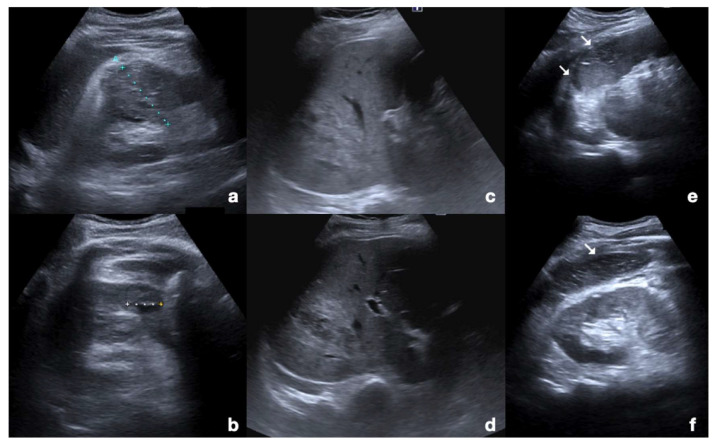
Non-enhanced US of parenchymal injuries. (**a**) Longitudinal view of the right kidney adequate to appreciate the renal parenchymal hematoma in its whole extension; (**b**) axial view of the same kidney showing another smaller hematoma. (**c**,**d**) Axial views of the liver showing a wide lacero-contusive area in the right lobe (arrows). (**e**,**f**) Multiple lacero-contusive areas of the spleen (arrows).

**Figure 2 diagnostics-12-00435-f002:**
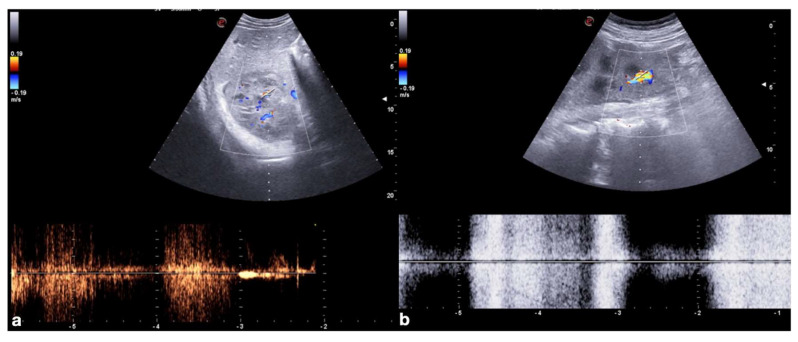
Role of color and spectral Doppler in the detection of post-traumatic vascular complications. Color (upper row) and spectral (bottom row) Doppler of the right kidney show post-traumatic pseudo-aneurysm in a 20−year-old man admitted to the emergency department for hematuria two weeks after a car accident and previous CT diagnosis of traumatic right kidney contusion (**a**); color Doppler shows turbulent flow in the false aneurysm, whereas spectral Doppler shows a “to and fro” spectrum. Color (upper row) and spectral (bottom row) Doppler of left kidney show post-traumatic arteriovenous fistula in a 45−year-old woman admitted at emergency department for penetrating injury (**b**); color Doppler shows aliasing artifact due to the presence of a focus of increased blood flow, then confirmed at pulsed Doppler that shows high-velocity peak without a clear diastolic flow.

**Figure 3 diagnostics-12-00435-f003:**
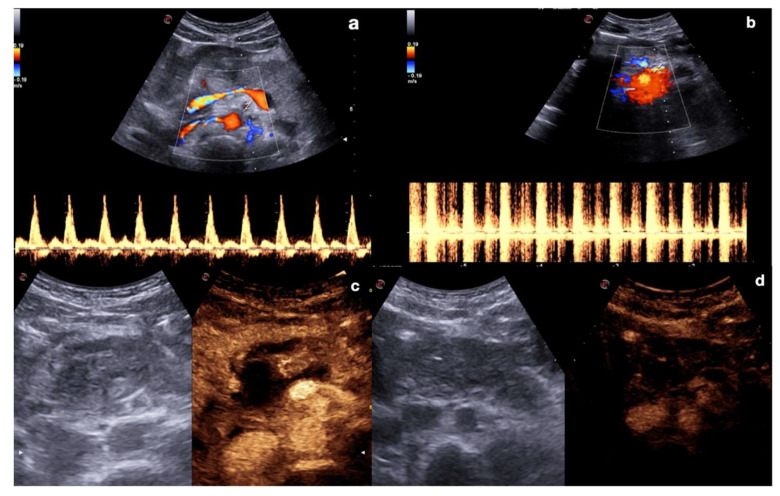
Artifacts in color and spectral Doppler after CEUS (**a**,**b**), and in repeated CEUS after Doppler (**c**,**d**). Mesenteric artery color and spectral Doppler before (**a**) and after (**b**) UCA administration showed alteration in the color map (upper row) as well as in the waveform (bottom row) evaluation due to the relative turbulence generated by the micro-bubbles within the vessel. Upper abdomen CEUS in pancreatic trauma before (**c**) and after (**d**) Doppler study shows relative destruction of the micro-bubbles due to high MI during the Doppler study, making the post-Doppler contrast study unable to visualize organs and tissues properly.

**Figure 4 diagnostics-12-00435-f004:**
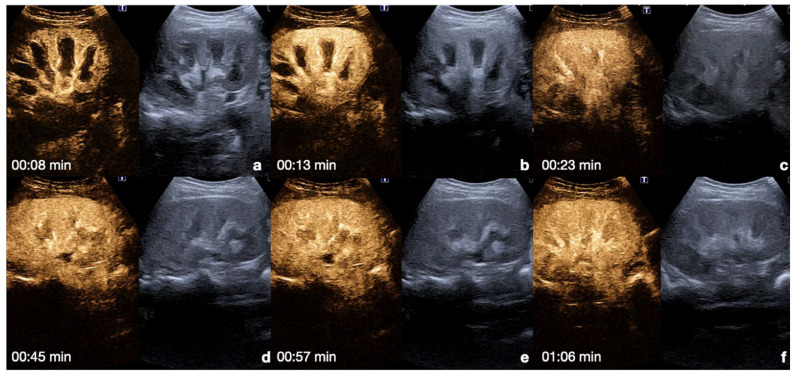
CEUS findings in a normal kidney (**a**–**f**). Note the progressive physiological enhancement of the cortex and the medulla. Adopted from ref. [30], 2021, Iacobellis, F.; et al.

**Figure 5 diagnostics-12-00435-f005:**
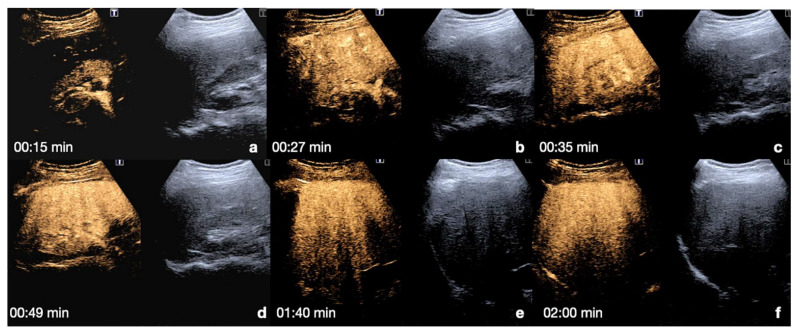
CEUS findings in a normal liver (**a**–**f**). Note the progressive physiological enhancement of the liver in the different phases. Adopted from ref. [30], 2021, Iacobellis, F.; et al.

**Figure 6 diagnostics-12-00435-f006:**
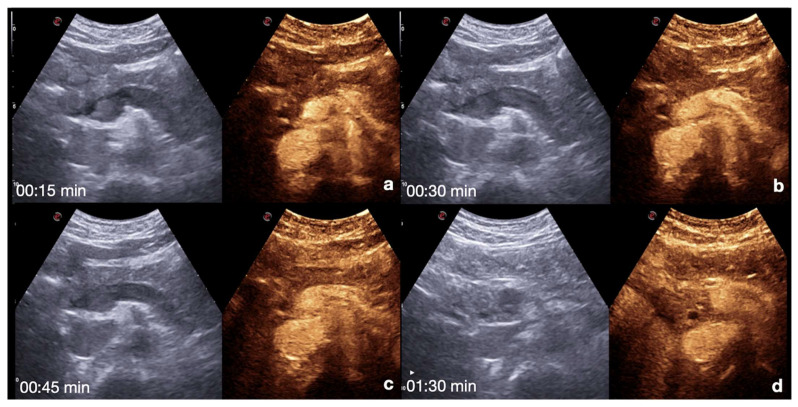
CEUS findings in a normal pancreas (**a**–**d**). Note the enhancement of the pancreas in the late arterial phase.

**Figure 7 diagnostics-12-00435-f007:**
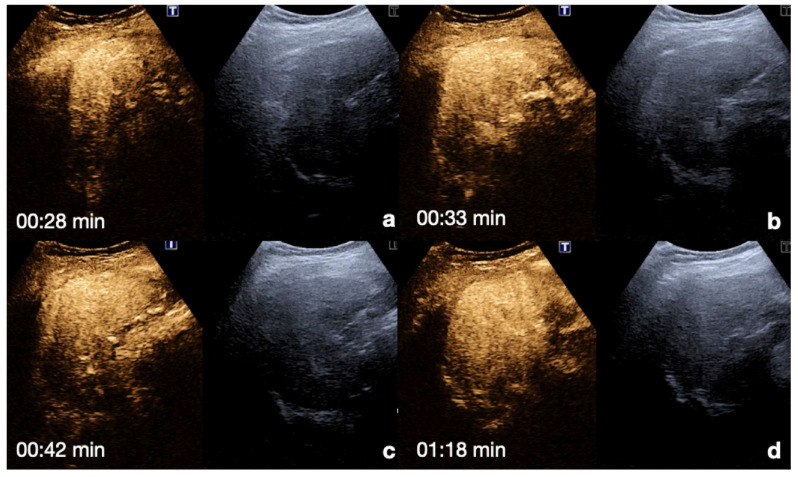
CEUS findings in a normal spleen (**a**–**d**). Note the progressive physiological enhancement after contrast medium injection. Adopted from ref. [30], 2021, Iacobellis, F.; et al.

**Figure 8 diagnostics-12-00435-f008:**
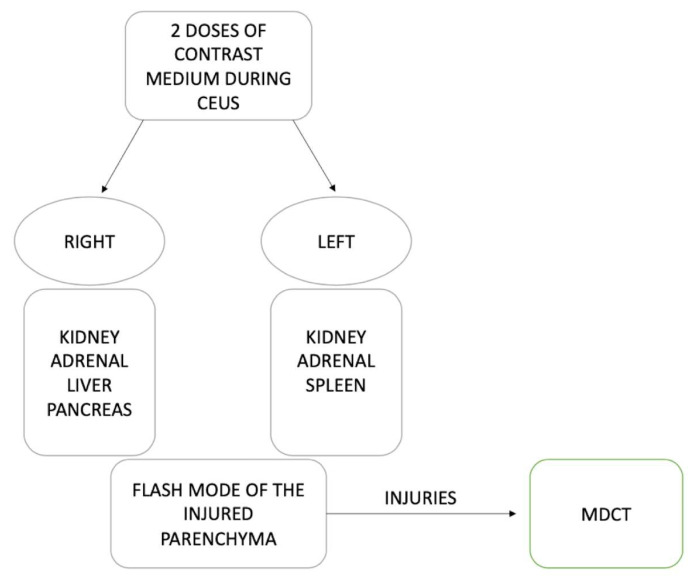
Low-grade trauma evaluated with CEUS.

**Figure 9 diagnostics-12-00435-f009:**
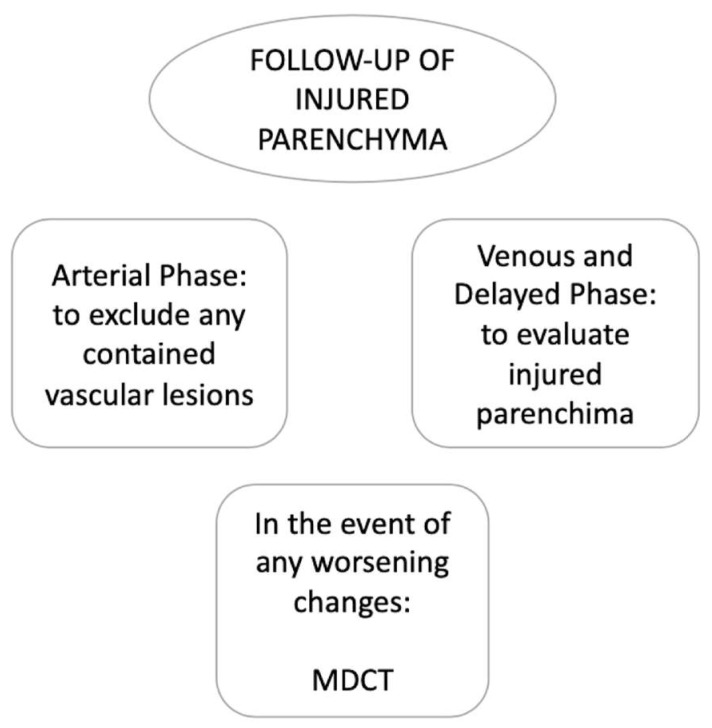
CEUS follow-up.

**Figure 10 diagnostics-12-00435-f010:**
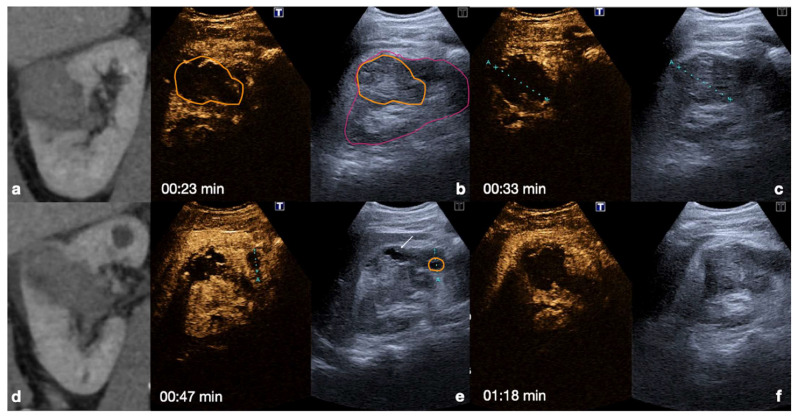
CT (**a**,**d**) and follow-up CEUS (**b**,**c**,**e**,**f**) of the right kidney in a 57−year-old patient who fell from a height. Follow-up CEUS was performed four days after the admission CT. Note at CEUS the progressive enhancement, at different time points, of the renal cortex in about 30 s (**b**,**c**) and the medulla, up to 2.5 min (**e**,**f**). The parenchymal hematomas appear as non-enhancing collections (**b**,**e**, orange lines) contained in the organ capsule (**b**, pink lines), without internal enhancing vessel or associated vascular injuries. Due to the physiological evolution of the hematoma, the follow-up, it shows small fluid anechoic areas related to the progressive resorption (**e**, arrow). Adopted from ref. [30], 2021, Iacobellis, F.; et al.

**Figure 11 diagnostics-12-00435-f011:**
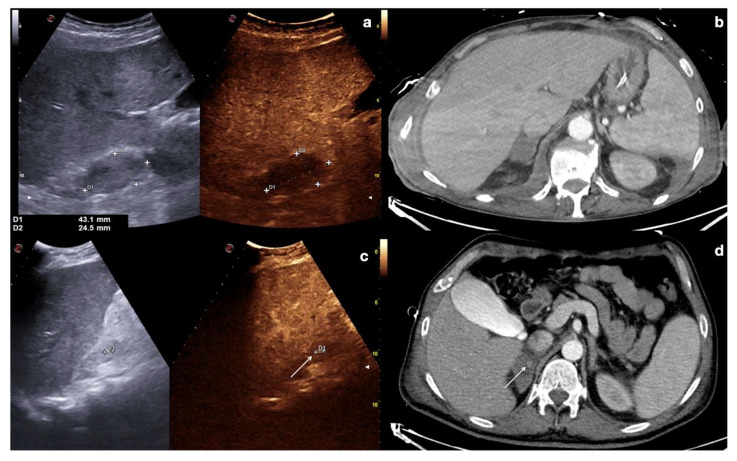
Adrenal gland hematoma. CEUS (**a**) and contrast-enhanced CT (**b**) of a 44−year-old male patient after a motor vehicle accident, showing right adrenal gland hematoma with no vascular complication. CEUS (**c**) and contrast-enhanced CT (**d**) follow up of a 36-year-old male patient on day 3 after a car accident; a pseudo-aneurysm within the right adrenal gland hematoma is visible (white arrows).

**Figure 12 diagnostics-12-00435-f012:**
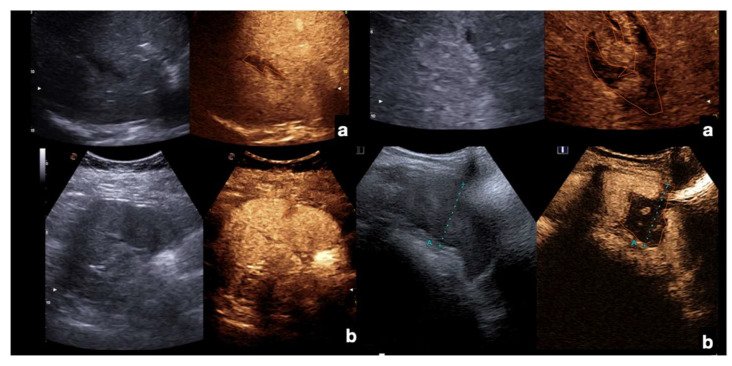
Example of CEUS imaging of low grade (first column) and high grade (second column) injuries in the liver (**a**) and spleen (**b**) after moderate energy blunt trauma. Orange lines indicate parenchymal lacerations/haematomas.

**Figure 13 diagnostics-12-00435-f013:**
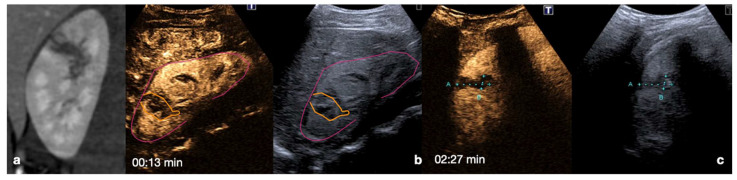
CT (**a**) and follow-up CEUS (**b**,**c**) of the right kidney in a 16−year-old patient with blunt trauma. At the admission, CT detected a kidney laceration reaching the organ capsule (**a**, arrow)**.** CEUS was performed ten days after trauma (**b**,**c**), showing minimal healing of the laceration without vascular injuries. Adopted from ref. [30], 2021, Iacobellis, F.; et al.

**Figure 14 diagnostics-12-00435-f014:**
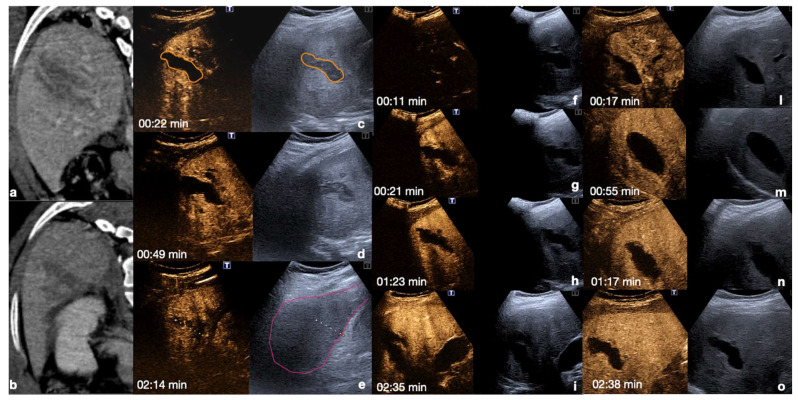
Admission CT (**a**,**b**) and follow-up CEUS (**c**–**o**) of a 35−year-old blunt trauma patient with multiple hepatic lacerations. Follow-up CEUS was performed 3 days (**c**–**e**), 11 days (**f**–**i**), and 20 days (**l**–**o**) after the admission CT. Note at CEUS the progressive enhancement of the liver parenchyma in the different phases. The parenchymal lacerations appear as non-enhancing bands (**c**, orange line), some of them reaching the liver capsule (pink line) (**e**,**i**). In the follow-up, it is important to look for possible vascular injuries (absent in this case) in the early arterial phase (**f**,**l**). Parenchymal lacerations appear progressively better demarcated, and more hypoechoic. Adopted from ref. [30], 2021, Iacobellis, F.; et al.

**Figure 15 diagnostics-12-00435-f015:**
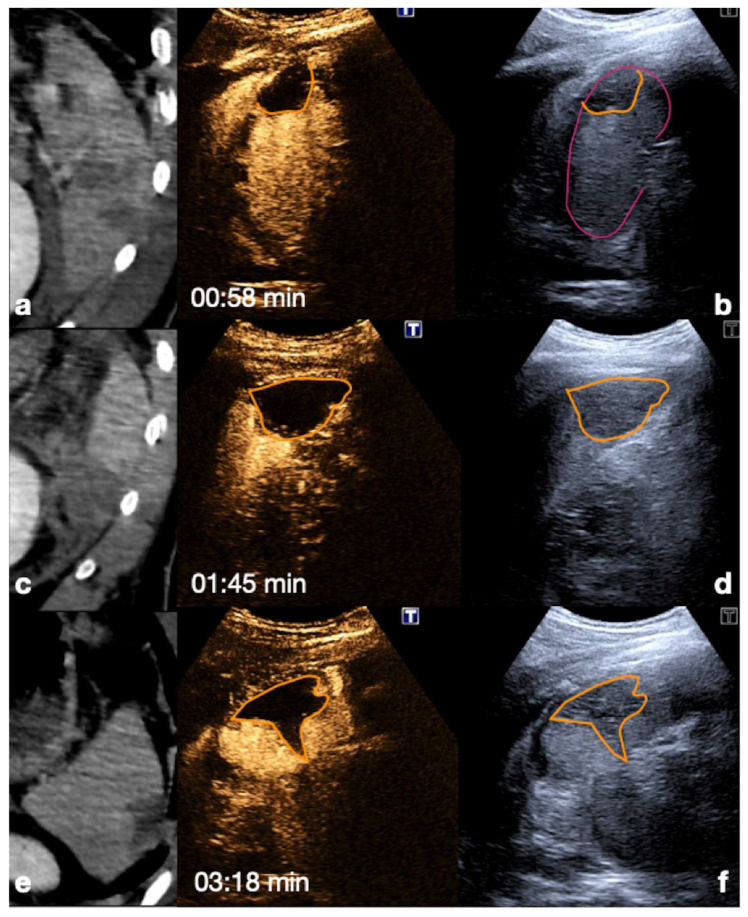
Admission CT (**a**,**c**,**e**) and follow-up CEUS (**b**,**d**,**f**) of the spleen in a 35−year-old patient after a fall from height. Follow-up CEUS was performed seven days after the CT. Note at CEUS the progressive enhancement, at different time points, of the healthy spleen parenchyma in venous phase, clearly demarcated from the subcapsular hematomas (**b**, orange line), from the contusion of the inferior-pole (**d**, orange line) and from a sub-capsular laceration (**f**, orange line). Adopted from ref. [30], 2021, Iacobellis, F.; et al.

**Figure 16 diagnostics-12-00435-f016:**
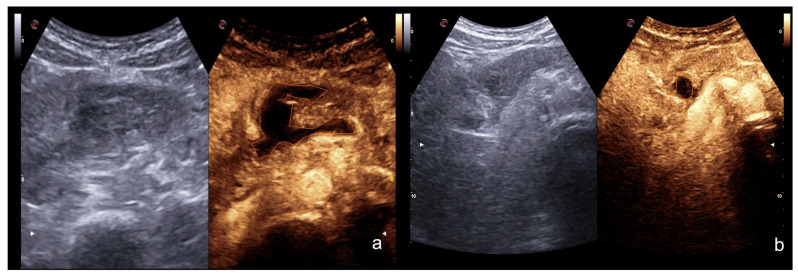
CEUS at day 1 (**a**) and after 2 weeks (**b**) in pancreatic trauma; note the reduction of the contusion area of pancreatic head, as well as of the peri-pancreatic fluid collection (orange line).

**Figure 17 diagnostics-12-00435-f017:**
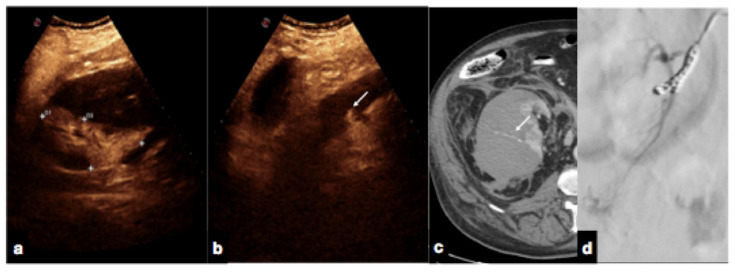
Follow-up CEUS of a 52−year-old blunt trauma patient with high-grade right kidney injury and extensive ischemia complication. Venous (**a**) phase CEUS examination shows multiple kidney lacerations with extensive ischemia complication and a small amount of perfused renal (caliper). At the same venous phase (**b**), CEUS active venous hemorrhage is well appreciated (arrow) confirmed at contrast-enhanced venous phase CT scan (**c,** arrow) and subsequent angiography (**d**).

**Figure 18 diagnostics-12-00435-f018:**
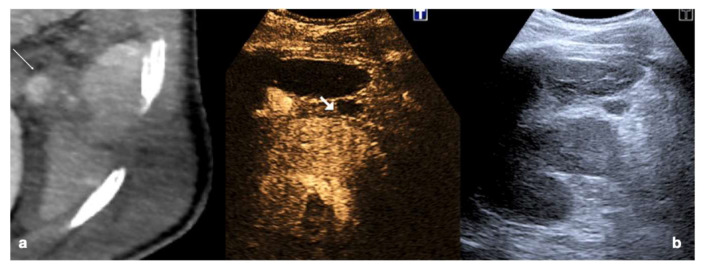
Admission CT (**a**) and follow-up CEUS (**b**) of the spleen of the same patient as Figure 14. At admission CT, in the arterial phase, was noticed a small hilar pseudoaneurysm (**a**, arrow). CEUS was performed after the embolization, showing the lack of vascular enhancement in the pseudoaneurysm site (**b**, arrow). Adopted from ref. [30], 2021, Iacobellis, F.; et al.

**Figure 19 diagnostics-12-00435-f019:**
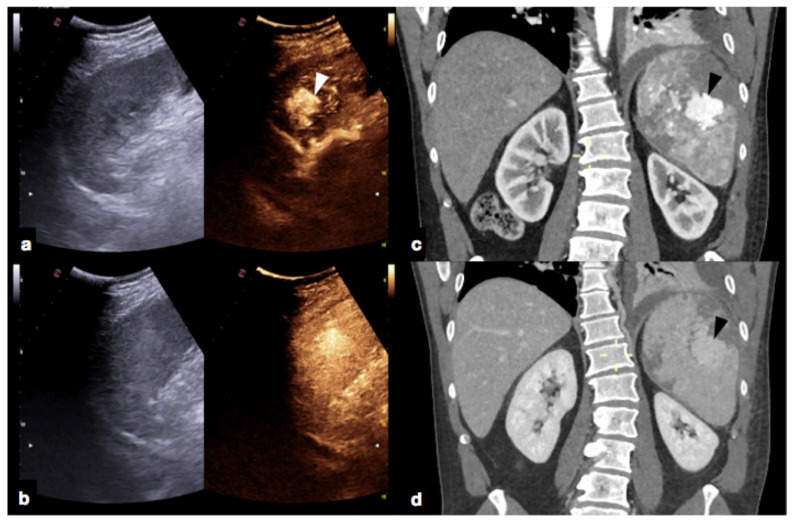
High grade traumatic splenic injury with vascular complication. Arterial (**a**) and venous (**b**) phase CEUS examination in a 25−year-old patient admitted at the emergency department for blunt abdominal trauma, showing multiple splenic lacerations and a voluminous arteriovenous fistula (white arrowhead). Subsequent arterial (**c**) and portal vein (**d**) phase contrast-enhanced CT scan further confirmed the diagnosis (black arrowhead).

**Figure 20 diagnostics-12-00435-f020:**
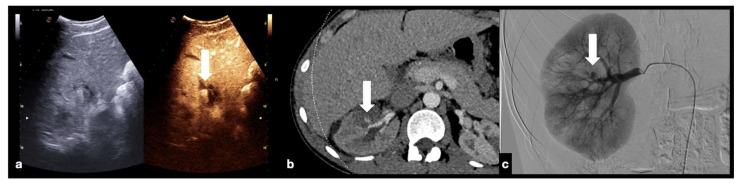
CEUS (**a**) of a 23−year-old man referring to the emergency department for direct blunt trauma to the right flank showed the presence of a small pseudoaneurysm (white arrow) inside the contusion area of the right kidney; the diagnosis was then confirmed at the axial arterial phase contrast-enhanced CT scan (**b**), as well as at angiography performed for treatment purposes (**c**).

**Figure 21 diagnostics-12-00435-f021:**
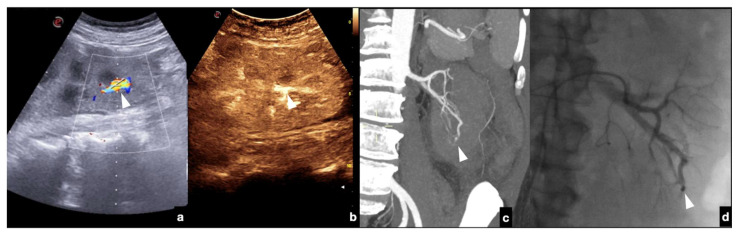
Companion case of Figure 2b. Renal arteriovenous fistula (white arrowhead) at color–Doppler US (**a**) and CEUS (**b**)**,** confirmed (**c**) at contrast-enhanced CT scan (arterial phase, coronal MIP reconstruction) and subsequent angiography (**d**).

**Figure 22 diagnostics-12-00435-f022:**
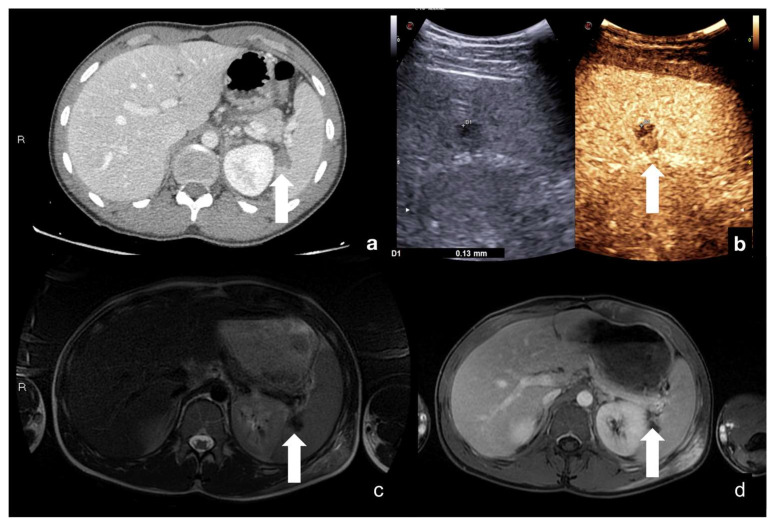
Multimodal evaluation of splenic laceration (white arrows) at portal phase CT scan (**a**), CEUS (**b**), fat-sat T2w (**c**), and post-contrast fat-sat T1w MRI (**d**), with good overlap of findings.

**Figure 23 diagnostics-12-00435-f023:**
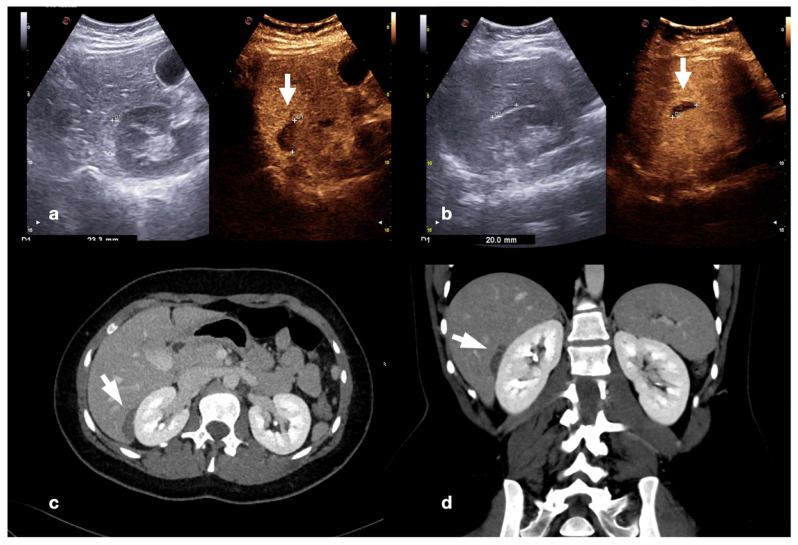
Sub-Glissonian hepatic hematoma in a 40−year-old woman. At B-mode US study (**a**,**b**, split image on the left), no definite hematomas was showed. At CEUS evaluation (**a**,**b**, split image on the right), the presence of a small sub-Glissonian hematoma was clearly delineated (arrow). Contrast-enhanced CT examination (**c**, axial and **d**, coronal view) confirmed the presence of the small non-bleeding sub-Glissonian hematoma (arrow).

**Figure 24 diagnostics-12-00435-f024:**
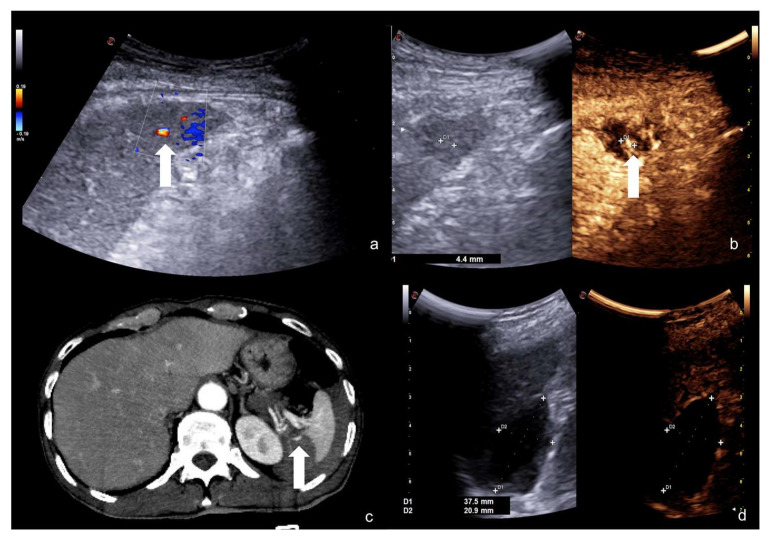
An example of multimodal visualization of post-traumatic splenic PSA (white arrows) at CD–US (**a**), CEUS (**b**), and arterial phase contrast-enhanced CT scan (**c**); CEUS follow-up examination after embolization (**d**) showed no evidence of residual PSA.

**Figure 25 diagnostics-12-00435-f025:**
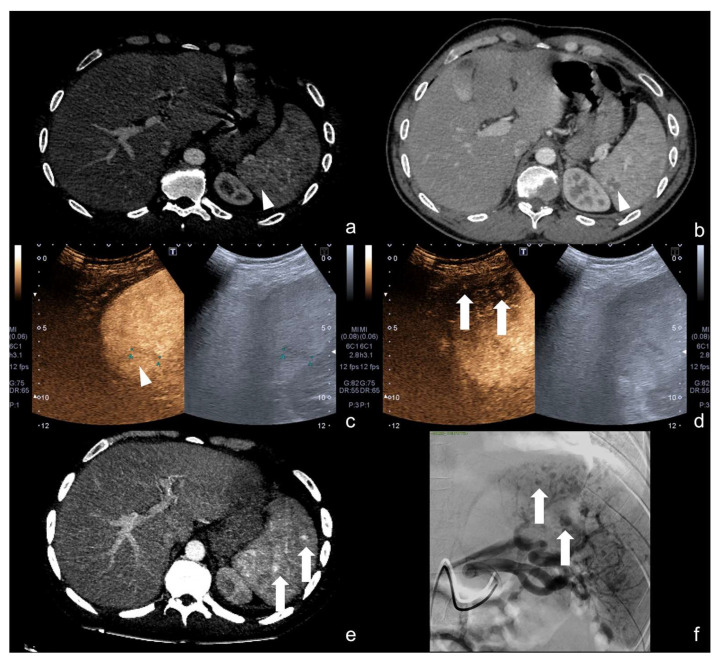
Splenic trauma in car accident with small splenic laceration visible on contrast-enhanced CT scan performed at emergency department (**a**,**b**, white arrowhead). CEUS examination performed 4 days after trauma confirmed the splenic laceration (**c**, white arrowhead); subsequent Flash mode CEUS (**d**) revealed multiple, small and diffuse intra-splenic PSAs (white arrows) not shown at admission arterial phase CT exam (**a**); these findings were confirmed at contrast enhanced CT scan (**e**, arrows) and angiography (**f**, arrows).

**Figure 26 diagnostics-12-00435-f026:**
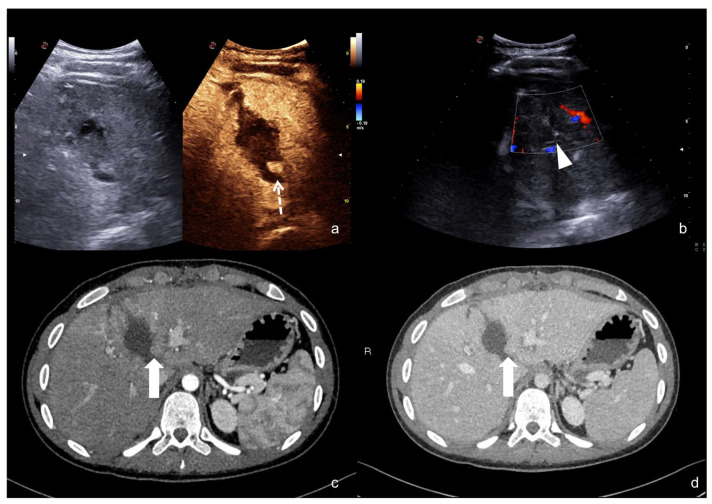
An example of possible pitfall at CEUS performed to follow-up a parenchymal laceration; in this case, a hepatic laceration. At CEUS examination, a nodular area of enhancement was visible within the hepatic laceration (**a**, dotted arrow), suspected for PSA; however, this finding was inconsistent with the preliminary CD–US evaluation, because the hepatic laceration area did not show any vascular pattern of PSA inside (**b**, white arrowhead). This finding was suspected for pseudo-nodular spared hepatic parenchyma and confirmed at biphasic contrast-enhanced CT examination (**c**, arterial phase, white arrow; **d**, venous phase, white arrow).

**Figure 27 diagnostics-12-00435-f027:**
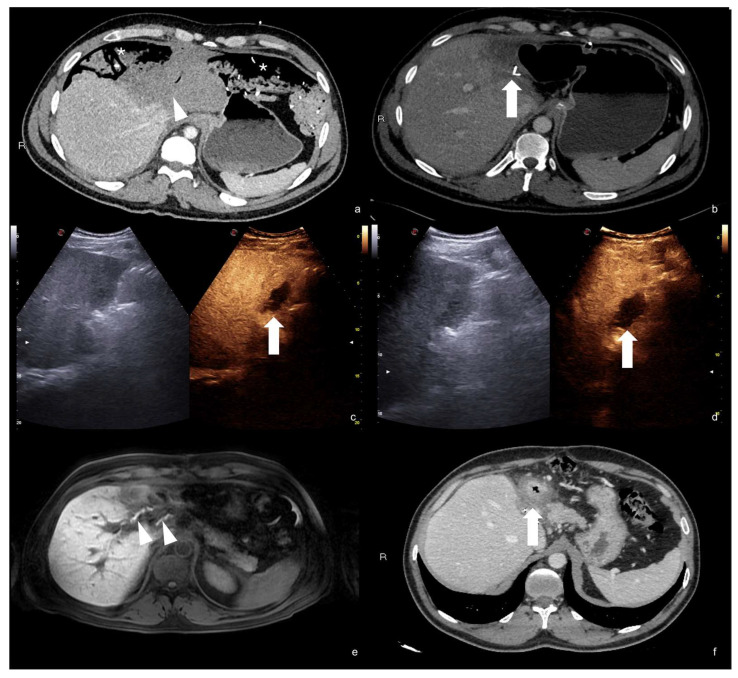
Post-traumatic laceration of the left hepatic lobe (**a**, white arrowhead) with evidence of surgical packing (asterisk); after surgical lobectomy, contrast-enhanced CT scan (**b**) showed a small fluid collection close to the biliary duct clipping (white arrow). Such collection increased in volume at follow-up CEUS examinations (**c**,**d**, respectively, 2 and 3 weeks after surgery, white arrows); this finding indirectly suggested a possible biliary leakage, although not directly viewable at CEUS (**c**,**d**, white arrows). This finding was confirmed at post-contrast hepatospecific phase fat-sat T1w MRI examination that showed biliary leak (**e**, arrowheads). Subsequent exclusion of the biliary leak with complete reabsorption of the biliary collection at CT examination (**f**, arrow).

**Table 1 diagnostics-12-00435-t001:** US scanning time after UCAs administration and flash mode technique.

Time and Flash Technique	Advantages
<0 s	Choose the best scan view in B-Mode and US Doppler studies (color, power, and pulse Doppler).
0 s	Injection of 2 mL of UCA followed by 10mL of saline solution.
10–20 s (early) 20–40 s (late)	Arterial phase: best depiction of contained vascular injuries, such as pseudoaneurysms and arteriovenous fistulas in the early phase.
2–6 min	Venous-late phases: distribution of the contrast in the whole organ. Best time to depict parenchymal injuries.
Flash mode	Destruction of bubbles and possibility to re-evaluate an area of interest.

**Table 2 diagnostics-12-00435-t002:** Enhancement characteristic for organ with the best opacification times.

Main Organs to Explore	Enhancement Characteristic
Kidney	Quick enhancement of the cortex after injection. Pyramids enhancement after 30 s. No excretory phase. Good evaluation up to 2.5 min.
Liver	Arterial phase: 10–40 s Hepatic and portal phases: 40–120 s Sinusoidal phase: 120–300 s Dual vascular supply permits homogeneous enhancement.
Pancreas	Arterial phase: 15–30 s Venous phase: 30–120 s The best moment to detect organ injury: venous phase.
Adrenal glands	Arterial phase: 20–40 s Homogeneous enhancement up to 5 min.
Spleen	Arterial phase: 12–20 s. Venous phase: 40–60 s up to 5–7 min. The best moment to detect organ injury: venous phase.

## Data Availability

Data sharing is not applicable.

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
