# Peer review of "The Technique and Advantages of Contrast-Enhanced Ultrasound in the Diagnosis and Follow-Up of Traumatic Abdomen Solid Organ Injuries"

_diagnostics, 2022, doi:10.3390/diagnostics12020435_

Round 1

Reviewer 1 Report

Review comments for the article submitted to Diagnostics, which is entitled “The technique and advantages of contrast-enhanced ultrasound in the diagnosis and follow-up of traumatic abdomen solid organ injuries”.

Overall, the topic is interesting to the readers of Diagnostics. The data provided by the authors is adequate to support their claim that the CEUS performs better than common US. And CEUS could potentially be an alternative for MDCT. I suggest its publication on Diagnostics after minor revisions. The comments are as follows:

  1. I would encourage the authors to cite more references in the introduction to provide a broader background for CEUS diagnosis, not only on traumatic abdomen solid organ injuries, but also on other diseases.
  2. Some of the writings requires polishing. For examples, (a) too many prepositions in single sentence. “Given the need to ... becoming more critical.” (b) the full names of the abbreviations “CEUS, US, MDCT” should be given when they firstly show up in the abstract.
  3. Is the SonoVue the only ultrasound contrast agent that is currently used in Europe? If not, the authors should revise the following sentence which has such implication. “SonoVue(TM) (Bracco, Milan, Italy) is the ultrasound contrast agent (UCA) currently used in Europe.”

Author Response

The Author would like to thank the Reviewer for their suggestions, useful to improve the manuscript quality

Overall, the topic is interesting to the readers of Diagnostics. The data provided by the authors is adequate to support their claim that the CEUS performs better than common US. And CEUS could potentially be an alternative for MDCT. I suggest its publication on Diagnostics after minor revisions. The comments are as follows:

  1. I would encourage the authors to cite more references in the introduction to provide a broader background for CEUS diagnosis, not only on traumatic abdomen solid organ injuries, but also on other diseases.
At the end of the introduction a sentence was added regarding other CEUS applications with related references.
  1. Some of the writings requires polishing. For examples, (a) too many prepositions in single sentence. “Given the need to ... becoming more critical.” (b) the full names of the abbreviations “CEUS, US, MDCT” should be given when they firstly show up in the abstract.
The sentence was modified as follow: Given the need to reduce exposure to ionizing radiation and to consider the risk of contrast-induced nephropathy associated with multiple administrations of iodinated contrast medium or single administrations in patients with kidney failure, the appropriate selection of trauma patients for MDCT is becoming more critical.  The abbreviations were added in the abstract, as suggested. The English language was reviewed by a native English speaker.
  1. Is the SonoVue the only ultrasound contrast agent that is currently used in Europe? If not, the authors should revise the following sentence which has such implication. “SonoVue(TM) (Bracco, Milan, Italy) is the ultrasound contrast agent (UCA) currently used in Europe.”
Actually SonoVue is the  ultrasound contrast agent used in Italy and in our experience, we have added the other two contrast agent used in Europe at the beginning of the dedicated paragraph: "The ultrasound contrast agent (UCA) currently approved in Europe are sulphur hexafluoride microbubbles (SonoVue(TM), Bracco, Milan, Italy) and perflutren microspheres (Optison, GE Healthcare(TM), and Luminity(TM), Lantheus Medical Imaging Massachusetts, United States). In our experience we adopt SonoVue…    Hoping these additions may have met your expectation, All our best regards The Authors    

Reviewer 2 Report

Dear authors the paper describes the technique for CEUS, as it has already been described in several previous papers.

From this standpoint it doesn't bring any news to the acutal knowldge. In fact the type of informations you give are available since at least a decade.

A similar paper has been published by Pinto F and Miele V in 2014.

It should be more interesting to know your clinical series experience with CEUS used during the follow up of NOm at your Institution

Could you please add your case series?

Author Response

The Author would like to thank the Reviewer for their suggestions, useful to improve the manuscript quality

Dear authors the paper describes the technique for CEUS, as it has already been described in several previous papers. 

From this standpoint it doesn't bring any news to the acutal knowldge. In fact the type of informations you give are available since at least a decade.

A similar paper has been published by Pinto F and Miele V in 2014. 

Dear Reviewer, actually, the aim of our review article is to underline the role and the management implication of the CEUS in trauma patients with solid organ injuries. It is not only focused on the technique but it examines also the advantages of CEUS in an integrated US examination starting with B-mode and Doppler evaluation and that proceed with CEUS in different clinical settings.    It should be more interesting to know your clinical series experience with CEUS used during the follow up of NOm at your Institution

Could you please add your case series?

The examination of our case series would go beyond the aim of the present article, conceived as pictorial review. However we have a broad, long-time experience in this field and for completeness we decided to further expand the illustrated series, also adding a case of pancreatic trauma (figure 16).     Hoping these additions may have met your expectation, All our best regards The Authors

Round 2

Reviewer 2 Report

Dear authors, the aim of you work was very clear to me, but as I stated in advance, there are plenty of papers about the technique and the advantages of CEUS in trauma setting.

I'm sorry but the paper doesn't add any novelty

Author Response

The Author would like to thank the Reviewer for their suggestions, useful to improve the manuscript qualit

It should be more interesting to know your clinical series experience with CEUS used during the follow up of NOm at your Institution   As previously requested, we have enriched the text with  step-by-step practical applications, tips and tricks during CEUS follow-up of NOM